# ^198^Au-Coated Superparamagnetic Iron Oxide Nanoparticles for Dual Magnetic Hyperthermia and Radionuclide Therapy of Hepatocellular Carcinoma

**DOI:** 10.3390/ijms24065282

**Published:** 2023-03-09

**Authors:** Nasrin Abbasi Gharibkandi, Michał Żuk, Fazilet Zumrut Biber Muftuler, Kamil Wawrowicz, Kinga Żelechowska-Matysiak, Aleksander Bilewicz

**Affiliations:** 1Centre of Radiochemistry and Nuclear Chemistry, Institute of Nuclear Chemistry and Technology, Dorodna 16 St., 03-195 Warsaw, Poland; 2Faculty of Chemistry, University of Warsaw, Pasteura 1 St., 02-093 Warsaw, Poland; 3Nuclear Applications Department, Institute of Nuclear Sciences, Ege University, 35040 Bornova, Izmir, Turkey

**Keywords:** core–shell nanoparticles, ^198^Au radionuclide, SPIONs, magnetic hyperthermia, radionuclide therapy

## Abstract

This study was performed to synthesize a radiopharmaceutical designed for multimodal hepatocellular carcinoma (HCC) treatment involving radionuclide therapy and magnetic hyperthermia. To achieve this goal, the superparamagnetic iron oxide (magnetite) nanoparticles (SPIONs) were covered with a layer of radioactive gold (^198^Au) creating core–shell nanoparticles (SPION@Au). The synthesized SPION@Au nanoparticles exhibited superparamagnetic properties with a saturation magnetization of 50 emu/g, which is lower than reported for uncoated SPIONs (83 emu/g). Nevertheless, the SPION@Au core–shell nanoparticles showed a sufficiently high saturation magnetization value which allows them to reach a temperature of 43 °C at a magnetic field frequency of 386 kHz. The cytotoxic effect of nonradioactive and radioactive SPION@Au–polyethylene glycol (PEG) bioconjugates was carried out by treating HepG2 cells with various concentrations (1.25–100.00 µg/mL) of the compound and radioactivity in range of 1.25–20 MBq/mL. The moderate cytotoxic effect of nonradioactive SPION@Au-PEG bioconjugates on HepG2 was observed. The cytotoxic effect associated with the β^−^ radiation emitted by ^198^Au was much greater and already reaches a cell survival fraction below 8% for 2.5 MBq/mL of radioactivity after 72 h. Thus, the killing of HepG2 cells in HCC therapy should be possible due to the combination of the heat-generating properties of the SPION-^198^Au–PEG conjugates and the radiotoxicity of the radiation emitted by ^198^Au.

## 1. Introduction

Hepatocellular carcinoma (HCC) is the most frequent primary liver malignant tumor which still has limited effective therapeutic protocols. Despite considerable progress in cancer therapy, such as targeted and immunotherapies, liver transplantation is still the best option to prolong the quality of life in patients with HCC. Unfortunately, most patients are in advanced stages and treatment is often challenging mostly due to late diagnosis and high tumor heterogeneity. Typically, chemotherapy and radiotherapy (most commonly with Sorafenib^®^) are given to patients with advanced HCC, but the overall therapeutic effect is unsatisfactory [1]. Ablation, transarterial chemoembolization, chemotherapy, and combination approaches are additional treatment methods for advanced HCC [2].

Transcatheter arterial chemoembolization (TACE) is currently used as the main therapy for unresectable HCC [3]. TACE is a locoregional treatment that prevents cancer growth by blocking a tumor’s blood supply, resulting in ischemic necrosis, and enabling simultaneous chemotherapeutics injection for additional chemotherapy. The direct application of this drug delivery system can help to reduce side effects and increase the therapeutic efficacy by limiting drug distribution to non-target tissues and boosting tumor drug levels [4]. This therapeutic approach was also improved by introducing drug-eluting beads (TACE-DEB) [5]. Radioembolization is another transarterial treatment for liver tumors. It consists of intra-arterial radiation injection using embolization carriers. This technology, similar to TACE, is based on the fact that liver tumors get blood from the hepatic artery rather than the portal vein. Hence, radioembolization enables preferential delivery of microscopic radiation particles and simultaneous embolization for the feeding artery. Currently, glass or resin microspheres with immobilized high-energy β^−^ radiation emitters such as ^90^Y, ^166^Ho, and ^188^Re are widely used [6].

Authors have recently proposed using nanotechnology to deliver chemotherapeutic drugs to HCC in several publications [7,8]. These works aimed to develop nanosystems ensuring maximum tumor vasculature penetration, optimal embolic activity, and effective delivery of a chemotherapeutic agent. Selectivity to cancer cells can be ensured by the utilization of the enhanced permeability and retention (EPR) effect. Due to the EPR effect, small-sized nanoparticle drugs can accumulate more effectively in the tumor than in healthy, normal tissues [9]. This phenomenon is possible due to the leaky tumor vasculature through which nanostructures can leave the bloodstream, pass through the gaps in the vessels’ endothelial lining, and enter the tumors. However, the combination of the embolization with chemotherapy by dispersion of drug-loaded nanoparticles in the embolization agent (such as lipiodol) is the best solution for TACE treatment [10]. Since iodized oil can be metabolized in healthy liver tissues but not in tumor tissues, it is frequently used in TACE. [11]. Heavy iodine atoms are also an ideal contrast agent that can be monitored with X-rays. A simple combination of drug solution and iodized oil is widely used in TACE [12]. However, this mixture has a quick release into the systemic circulation. Therefore, using drug-eluting nanoparticle emulsions in lipiodol significantly increases the effectiveness of the TACE therapy while reducing side effects. The results of studies using nanoparticles as radionuclide carriers for HCC therapy, apart from those employing lipiodol emulsion labeled with ^131^I [13] and ^188^Re complex incorporated into lipiodol, were not reported.

Arterial embolization hyperthermia (AEH) is another HCC treatment approach using nanotechnology [14]. It is a nanotechnology-based thermal therapy that exploits energy dissipation (heat) from forced magnetic hysteresis of a magnetite emulsion. This method combines the embolization process with hyperthermia and ensures a high concentration of superparamagnetic iron oxide-based nanoparticles (SPIONs) in HCC. Typically, SPION emulsions in lipiodol or its derivatives are used for therapeutic processes. Temperatures in the range of 41–45 °C were obtained during previously published studies [15]. Recently it has been discovered that external ionizing radiation in combination with magnetic hyperthermia has a significantly greater (synergetic) effect on killing various cancer cells than using the two methods separately [16]. The biological explanation for this effect is that mild hyperthermia increases intratumoral blood flow, and subsequent re-oxygenation makes it easier for ionizing radiation to generate reactive oxygen species (ROS) [17]. As a result, DNA damage increases. A similar effect was also achieved in magnetic hyperthermia combined with radionuclide therapy [18,19,20]. Recently Anilkumar et al. and Shivanna et al. reported that simultaneous induction of alternating magnetic fields and near-infrared laser on surface-functionalized SPIONs can increase hyperthermal temperature (43 °C) at lower SPION concentrations [21,22].

In the present work, we propose a new solution that involves using multimodal SPIONs, coated with β^−^ emitting radionuclide, allowing for simultaneous direct irradiation and local tumor hyperthermia. The main subject of this study is the synthesis of multifunctional SPIONs coated with a gold (^198^Au) layer to create core–shell nanoparticles (SPION@^198^Au) and the evaluation of their in vitro behavior. ^198^Au is a median energy β^−^ emitter that can be obtained with good specific activity by thermal neutron irradiation of the monoisotopic target ^197^Au. The therapeutic effect of SPION@^198^Au core–shell nanoparticles on HCC cells will be based on the simultaneous action of β^−^ radiation emitted by ^198^Au and magnetic hyperthermia generated on the SPION using an alternating external magnetic field. The concept of the studies is presented on Figure 1.

## 2. Results and Discussion

### 2.1. Synthesis and Characterization of Fe_3_O_4_@Au and Radioactive Fe_3_O_4_@Au Core–Shell Nanoparticles

The Fe_3_O_4_@Au core–shell nanoparticles were successfully prepared by the coprecipitation method with subsequent reduction of Au^3+^ cations with citrate on the nanoparticle surface. As previously reported [20], coating efficacy yielded greater than 99%. Transmission electron microscopy (TEM) studies (Figure 2) revealed that the synthesized SPIONs-citrate, SPION@Au, and SPION@Au–polyethylene glycol (PEG) nanoparticles were generally spherical with 12–17 nm diameters and apparent aggregation in the vacuum environment of TEM. The Au which covered the SPIONs (Figure 2b) made them darker, while some nanoparticles remained uncovered. This might be due to the non-uniform deposition of Au on some parts of nanoparticles, leaving some Fe_3_O_4_ cores exposed. Non-uniform distribution is a natural outcome, especially when spherical nanoparticles are considered. Usually, this type of nanomaterial’s size is characterized by a Gaussian distribution. Thus, it is a typical effect, and therefore homogenous shell formation control is difficult to achieve. It must be emphasized that this will have no negative impact on further therapeutic efficacy investigations, as deposition of Au on the SPION surface was reproducible across all of the syntheses performed during these studies. Thus, the investigated conjugate maintained similar characteristics. Figure 2a shows a visible organic shell (crown) made of citrate, but as shown in Figure 2b, removal of the citrate is observed when the surface is covered with a layer of Au. Figure 2c shows the Fe_3_O_4_@Au core–shell nanoparticles additionally coated with PEG. The synthesis of PEG-nanoparticle bioconjugates was performed using the known affinity of thiol groups to the surface of noble metals, e.g., Au [23]. HS-PEG-COOH with an average molecular weight of 5000 Da was used to enhance nanoparticle dispersity and stabilization. TEM images of the SPION@Au-PEG (Figure 2c) also revealed the presence of an organic shell with high electron permeability. The linked PEG molecules are the root cause of it. During the synthesis, colloidal Au nanoparticles are also formed, but the magnetic separation of the product makes it impossible to see the individual Au nanoparticles.

Dynamic light scattering (DLS) analysis was used to monitor the hydrodynamic size distribution of SPION suspensions at each step. The average hydrodynamic diameter and zeta potential of the SPION@Au and SPION@Au–PEG nanoparticles are shown in Table 1.

The presence of the water-solvation layer reveals why the DLS data, as shown in Table 1, indicate a much greater diameter than the TEM analysis. Surprisingly, the hydrodynamic diameters of both SPION@Au and SPION@Au–PEG are many times greater than the corresponding values for Au and Au–PEG nanoparticles, which are 19.3 nm and 21.6 nm for Au and Au–PEG nanoparticles, respectively, for Au nanoparticles 15 nm in size according to TEM [24]. Therefore, in the case of SPION@Au and SPION@Au–PEG, we observe in the solution agglomerates of five or six nanoparticles. Moreover, the stability of the suspension was estimated via zeta potential measurements. Due to the repulsive Coulomb forces, the negative values of this potential demonstrate the remarkable stability of the nanoparticle suspension.

The stability of the SPION@Au–PEG suspension was measured in both deionized (DI) water and a 0.9% NaCl solution. The nanoparticles showed superb stability during the 14 day test period. Additionally, due to the noble nature of metallic gold, the release of Au in ionic form is not observed. The radioactive SPION-^198^Au–PEG conjugates were obtained in the same way as the nonradioactive ones, except for using ^198^Au dissolved in aqua regia. ^198^Au was achieved using a significant specific activity through thermal neutron irradiation of the Au target. The high cross-section for ^197^Au(n,ɣ)^198^Au nuclear reaction (98.7 barns) makes it possible to receive 16 GBq of ^198^Au in the reactor Maria, Świerk (Poland) (1 mg Au target, 10^14^n/cm^2^/s thermal neutron flux, 70 h irradiation). Due to auspicious nuclear characteristics (t_1/2_ = 2.7 d, β_max_ = 0.96 MeV), ^198^Au seems to be a promising low energy β^−^ emitter for targeted radionuclide therapies, especially the treatment of small inoperable tumors. Unfortunately, neither the Au^+^ nor Au^3+^ cations can form stable complexes. Therefore, it is impossible to attach them to biomolecules via chelating agents commonly used in nuclear medicine. Thanks to the considerable development in nanotechnology, Au atoms in the form of nanoparticles can be considered a practical radioactivity carrier that could be introduced safely to the human body [25]. Therefore, using ^198^Au emitting ionizing radiation as a shell on iron oxide nanoparticles delivering hyperthermia could make this multimodal platform practical for combination therapy.

### 2.2. Magnetic Properties of SPION@Au

The magnetic characteristics of SPION@Au were measured using a quantum dot (QD) vibrating sample magnetometer (VSM) with magnetic field strength (H) ranging from −2.0 T to +2.0 T at temperatures of 100 K and 300 K (as seen in Figure 3).

The absence of a coercive field on the magnetization curve reveals the superparamagnetic properties of the generated nanoparticles. The magnetization rises dramatically up to ~1000 Oe and the saturation point is ~50 emu/g, which is lower than saturation points reported in publications for uncoated SPIONs (83 emu/g) [26]. Nevertheless, the SPION@Au core–shell nanoparticles have high enough saturation magnetization values for their possible application in magnetic hyperthermia. The saturation magnetization value is also not affected by further modification with PEG-type organic molecules or biomolecules, as we have demonstrated in our previous studies [27,28].

Specific absorption rate (SAR), referring to the power P [W] produced per mass m [g] of nanoparticles, is the most commonly cited parameter for estimating the heat conversion efficacy of samples. The SAR due to heating in a magnetic field can be defined as follows:(1)$$\mathrm{SAR}=\frac{d\cdot C}{N_{p}}{\left(\frac{\mathrm{dT}}{\mathrm{dt}}\right)}_{\max}$$
where d is dispersant density (kg/m^3^), C is the specific thermal conductivity of the media (in the present case, water, C = 4180 J/kg K), N_p_ is nanoparticle concentration (kg/m^3^), T represents the temperature (K), and t is the time (s).

The magnetic field H [kA/m] and its frequency f [Hz] can both affect the SAR parameter. The dependence of SAR on the magnetic field frequency for SPION@Au–PEG is shown in Figure 4.

According to the practical application of magnetic hyperthermia in cancer therapy, the optimal magnetic field frequency should vary between 345 and 488 kHz [29]. The SAR values for SPION@Au nanoparticles at these frequencies are 200–320 W/g, which are adequate to induce magnetic hyperthermia. The minimum concentration of SPION@Au–PEG nanoparticles required to achieve a temperature >42 °C was determined at the frequency of 386 kHz. Figure 5 shows the heating rates of SPION a, coated with citrate, b, SPION@Au, and c, SPION@Au–PEG nanoparticles. The presented time dependencies demonstrate that the coating of SPION–citrate with the Au layer slightly improves the heat generation capability of nanoparticles. However, the attachment of long PEG chains considerably lessens the SAR value, probably because of changes in the hydrodynamic diameter and, thus Brownian losses as well as changes in the saturation magnetization value [30]. Nevertheless, as shown in Figure 5a concentration of SPION@Au–PEG nanoparticles greater than 6 mg/mL enables the quick achievement of the temperature required for cell apoptosis or necrosis.

### 2.3. In Vitro Cytotoxicity Assay

The cytotoxic effect of nonradioactive SPION@Au–PEG bioconjugates was conducted using various concentrations (0–100.00 µg/mL) of the compound for 24, 48, and 72 h. The metabolic activity of the cells was evaluated using the MTS assay (Figure 6). Based on cytotoxicity studies conducted on nonradioactive compounds, SPION@Au–PEG nanoparticles were not toxic to the HepG2 cells until the concentration reached 12.5 µg/mL. Cell viability decreased gradually with increased compound concentrations, eventually reaching about 35% viability at the remarkably high concentration of 100 µg/mL. Similar cytotoxicity results were obtained in the studies of SPION nanoclusters coated with 2 kDa N-Alkyl-polyethyleneimine in the concentration range of 0–20 ug/mL [31]. The obtained cytotoxic effects of SPION@Au–PEG bioconjugates on HepG2 cells may be related to the limited release of Fe^2+^/Fe^3+^ ions from the magnetite nanoparticles. Since HepG2 cells contain a significant concentration of H_2_O_2_ [32], Fe^2+^ ions can participate in the generation of extremely hazardous ROS in the Fenton reaction:H_2_O_2_ + Fe^2+^ → OH^•^ + OH^−^ + Fe^3+^(2)

It is well known from the literature that Fe_3_O_4_ nanoparticles can generate small amounts of ROS [33,34]. Additionally, it has been recently shown that ROS generation can be enhanced with the application of alternating magnetic fields without a measurable temperature rise [35,36]. Under neutral conditions, Voinov et al. have demonstrated that γ-Fe_2_O_3_ nanoparticles produce hydroxyl radicals primarily on the surface rather than dissolution of free ions [37]. At lower pH, such as the microenvironment of a lysosome, iron ions can be released from the nanoparticle surface resulting in a greater extent of homogeneous catalysis [38]. Since the Au layering of SPION is not complete, SPION@Au nanoparticles can also generate ROS. Such an effect, to a lesser extent, was also observed in our previous toxicity studies of SPION@Au–PEG bioconjugates on ovarian cancer cells (SKOV-3) which also show elevated levels of H_2_O_2_ in the cytoplasm [20].

The substantially more intense cytotoxic effect was found once radioactive SPION-^198^Au–PEG conjugates interacted with HepG2 cells. The dependence of cell viability on the activity of SPIONs-^198^Au–PEG is shown in Figure 7.

As presented in Figure 7, we observed toxicity in a dose-dependent manner progressing over time. After 24 h of treatment, no decrease in cell viability was observed in any of the tested activity concentrations. Significant inhibition of cell metabolic activity was found primarily after 48 h even when the lowest concentration (1.25 MBq/mL) of radioconjugate was applied. The survival ratio decreased with increased radioconjugate concentration and reached 25% in the highest dose (20 MBq/mL). Interestingly, after 72 h, the cell survival fraction was below 5% which directly proves that changes induced by synthesized conjugates progress over time. Comparing Figure 6 and Figure 7, it is clear that the cytotoxic effect through the chemical generation of ROS is negligible compared to the radiotoxic effect presented in Figure 7. The presented results are consistent with the results published by Wu et al. [39] investigating the effect of the other β^−^ emitter, ^177^Lu on HepG2 cells. As in our studies, for 1.85 MBq activity of free ^177^Lu, cell survival after 24 h was ~85%.

## 3. Materials and Methods

### 3.1. Materials

Iron (III) chloride hexahydrate (97%), iron (II) chloride tetrahydrate (>99%), triammonium citrate dihydrate (>97.0%), glycine (>99.0%), and bifunctional thiolated carboxylic-polyethylene 4α;-sulfanyl-ω carboxy PEG HS-PEG-COOH, 5 kDa were purchased from Sigma-Aldrich (St. Louis, MO, USA). Sodium hydroxide (>99.9%) and 25% ammonia solution were supplied from POCH, Wrocław, Poland. PBS 10x buffer (pH 7.4) was purchased from VWR Life Science (Randor, PA, USA). MTS reagent (3-[4,5,dimethylthiazol-2-yl]-5-[3-carboxymethoxy-phenyl]-2-[4-sulfophenyl]-2H-tetrazolium, inner salt) was supplied from Promega (Promega, Madison, WI, USA). HepG2 cells were obtained from the American Type Tissue Culture Collection (ATCC, Rockville, MD, USA) and cultured according to the ATCC protocol. ^198^Au was obtained by neutron irradiation of a natural Au target (99.99%) at research reactor Maria in Świerk, Poland). The neutron flux was 1.5 × 10^14^ n cm^−2^ s^−1^ and the irradiation time was 1 h. All solutions were prepared using ultrapure DI water (18.2 MΩ·cm, Hydrolab, Straszyn, Poland).

### 3.2. Techniques

The size and morphology of nanoparticles were examined using a Zeiss Libra 120 Plus TEM operating at 120 kV (Zeiss, Stuttgart, Germany). The DLS method was used to analyze the hydrodynamic size of the synthesized nanoparticles and PEG conjugates. The hydrodynamic diameter and zeta potential measurements were conducted in 1 mM PBS pH 7.4 buffer using a Zetasizer Nano ZS (Malvern Panalytical, Malvern, Worcestershire, UK). The magnetic properties of the obtained nanoparticles were tested using a QD VSM vibrating magnetometer by NanoMagnetics Instruments (Oxford, UK) operating in the range of –2.0 to +2.0 T. Measurements were carried out in the temperature range from 100–300 K with an accuracy of 0.01 K. The SAR values were determined using MaNIaC Controller software supplied with D5 series equipment. The MTS assay absorbance values were evaluated at 490 nm via an Apollo 11LB913 microplate reader (Berthold, Bad Wildbad, Germany). The radioactivity of samples was measured using Wizard^®^ 2 automatic gamma counter (Perkin Elmer, Waltham, MA, USA).

### 3.3. Synthesis of Fe_3_O_4_ Nanoparticles

Fe_3_O_4_ nanoparticles were generated by mixing 0.2 M FeCl_3_ and 0.1 M FeCl_2_ solutions in a 2:1 volume ratio and coprecipitation with 25% NH_3(aq)_ solution at 95 °C for 15 min. After that, 300 µL of 0.46 M triammonium citrate dihydrate was introduced to stabilize the core. The reaction was then continued for another 30 min at the previous settings. The product was separated with a solid magnet and washed with cold acetone twice.

### 3.4. Synthesis of Fe_3_O_4_@Au Core–Shell Nanoparticles

SPION@Au core–shell nanoparticles were synthesized using the procedure elaborated by Zhou et al. [40]. Briefly, 4 mg of AuCl_3_x3H_2_O containing 2 mg of elemental Au was dissolved in 200 µL of DI water and then added to 250 µL of 20 mg/mL SPION@citrate suspension solution containing ~5 mg of Fe_3_O_4_. The coating reaction was allowed to run for 30 min with moderate stirring (600 rpm) at 100 °C. The product was magnetically separated and repeatedly washed with DI water. The obtained core–shell nanoparticles were combined with HS-PEG-COOH, 5 kDa in a 1:1 mass ratio, in 10 mM PBS solution for 2 h at room temperature.

The radioactive Fe_3_O_4_@^198^Au core–shell nanoparticles were obtained using an identical procedure. The only difference was that nonradioactive AuCl_3_ was used in place of ^198^AuCl_3_ obtained by dissolution of neutron-irradiated Au foil.

### 3.5. Synthesis of SPION@Au-PEG Nanoparticles

PEG (5000 kDa) was used to stabilize the nanoparticles in aqueous media. 2.5 mg of solid HS-PEG-COOH, 30 µL of 0.1 M PBS (pH 7.4), 20 µL of DI water, and 250 µL of SPION@Au solution containing 2.5 mg of Fe_3_O_4_ were added to the reaction tube. The solution was finally mixed overnight at room temperature. The obtained product was separated using a magnet and washed two times with water.

### 3.6. Synthesis of Radioactive SPION@^198^Au-PEG Nanoparticles

^198^Au was produced by the neutron irradiation of 4 mg of metallic ^197^Au target in the nuclear reactor core for 1 h with a 1.5 × 10^14^ n cm^−2^ s^−1^ neutron flux. The irradiated Au target was dissolved in 200 µL of aqua regia at 130 °C and evaporated. Then, 200 µL of 0.05 M HCl was added and evaporated three times before repeating the procedure with water. ^198^Au shells around SPIONs were obtained using the procedure described above for nonradioactive SPION@Au nanoparticle synthesis. The same pegylation procedure as in point 3.5 was also used.

### 3.7. Stability Studies

Following synthesis, the SPION@Au-PEG nanoparticles were separated from the reaction mixture and suspended in DI water and 0.9% NaCl solution for 14 days. The aggregation tendency was characterized by the measurements of their hydrodynamic diameter using DLS.

### 3.8. Determination of the SAR

To evaluate the heat generation ability of the obtained samples in an alternating magnetic field, the aqueous suspension samples of SPION@Au-PEG were implanted in a thermostated copper coil to record temperature changes as a result of the alternating magnetic field at frequencies at 163–633 kHz and an amplitude of 23 kA/m. The measurements were carried out until the temperature reached 55 °C or 5 min elapsed. The SAR values were determined by the ZaR subprogram of MaNIaC 1.0 Software.

### 3.9. In Vitro Cytotoxicity Studies

The cytotoxicity studies were performed by MTS assay for nonradioactive Fe_3_O_4_@Au-PEG and radioactive Fe_3_O_4_@^198^Au-PEG core–shell nanoparticles from 0.0 to 100 µg/mL and radioactivity from 0 to 20 MBq/mL. In the case of radioactive nanoparticles, the radioactivity of ^198^Au administered to the cells was regulated by changing the concentration of the ^198^Au–PEG conjugate. We prepared a solution of ^198^Au–PEG with a concentration of 1 MBq/µg and by increasing the concentration we obtained solutions with higher radioactivity concentrations, up to 20 MBq/mL.

HepG2 cells were cultured and seeded one day before the experiment in 96-well plates at a cell density of 2.5 × 10^3^ per well. Then, the cells were washed with PBS and treated with increasing concentrations of the studied compounds. Treated cells were incubated overnight at 37 °C in a 5% CO_2_ atmosphere. After this time, the cells were gently washed twice with PBS, and then fresh media with the tested compounds was added. Then, the plates were incubated for 24, 48, and 72 h. After these incubation points, the media was removed and fresh media followed by MTS reagent were added to each well, and the incubation continued for an additional 2 h at 37 °C and 5% CO_2_ atmosphere in the dark. Lastly, the absorbance was measured at 490 nm with a microplate reader.

## 4. Conclusions

SPIONs coated with radioactive Au, ^198^Au, and modified with PEG were synthesized as promising agents for combined targeted radionuclide therapy and magnetic hyperthermia of HCC. The PEG coating can prevent nanoparticle aggregation, promote biocompatibility, and allow for nanoparticle dispersion in the embolization agent (such as lipiodol) to form an emulsion for arterial radioembolization and hyperthermia. Even though the saturation magnetization values of SPION@Au–PEG nanoparticles are lower compared to uncoated SPIONs, the heating abilities of the nanoparticles were maintained, making it possible to reach the mild hyperthermia temperatures and make them acceptable for guided delivery to the target tissues in an external magnetic field. In addition, HepG2 cells exhibit a robust radiotoxic response to the radioactive SPION-^198^Au-PEG conjugates, specifically after 72 h. Hence, the destruction of HepG2 cells in HCC therapy should be possible by the combination of the heat generation properties of the SPION-^198^Au-PEG conjugates and the radiotoxic qualities of the radiation generated by ^198^Au. Incorporating this therapy into practice should be more straightforward as magnetic hyperthermia therapy of HCC using SPION dispersed in lipiodol has already been developed [41]. However, improvement of this method by introducing multimodality may become an interesting alternative, especially when such an aggressive and diversified disease as HCC is considered.

## Figures and Tables

**Figure 1 ijms-24-05282-f001:**
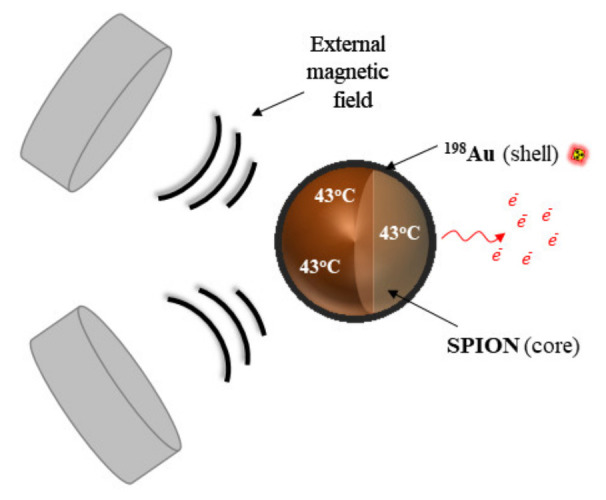
Investigated concept of application of SPIONs coated with a ^198^Au layer for simultaneous treatment of HCC by β^−^ radiation emitted by ^198^Au and magnetic hyperthermia generated on SPIONs.

**Figure 2 ijms-24-05282-f002:**
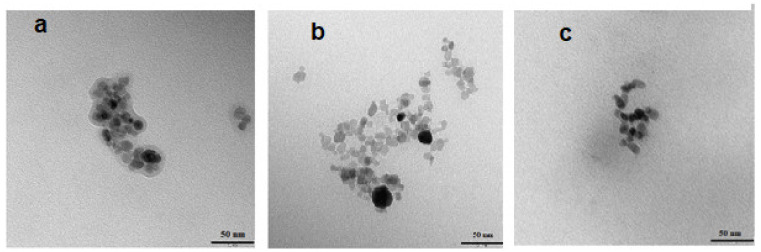
TEM images of synthesized SPIONs citrate (**a**), SPION@Au (**b**), and SPIONs@Au–PEG (**c**).

**Figure 3 ijms-24-05282-f003:**
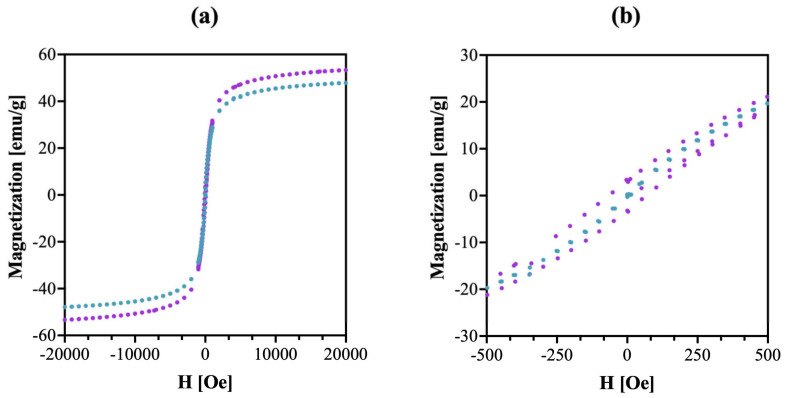
Magnetization isotherm for SPION@Au nanoparticles. Full hysteresis (**a**), fragment −500–500 Oe, (**b**). **……** 100 K, **…… **300 K.

**Figure 4 ijms-24-05282-f004:**
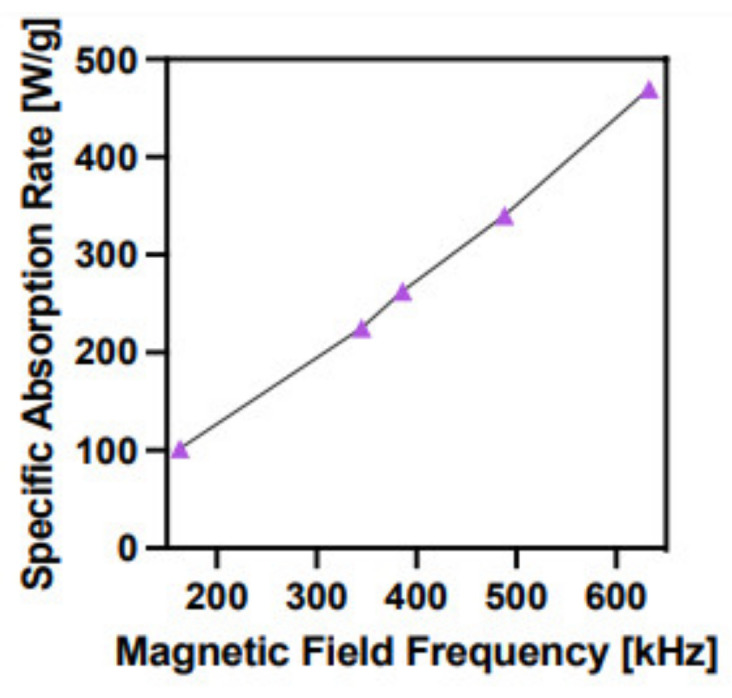
Dependence of SAR for SPION@Au nanoparticles on the magnetic field frequency.

**Figure 5 ijms-24-05282-f005:**
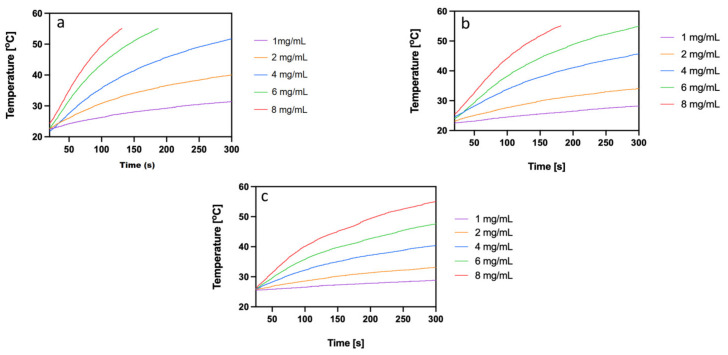
Heating of SPION: (**a**) coated with citrate, (**b**) SPION@Au, and (**c**) SPION@Au–PEG nanoparticles in alternating magnetic fields.

**Figure 6 ijms-24-05282-f006:**
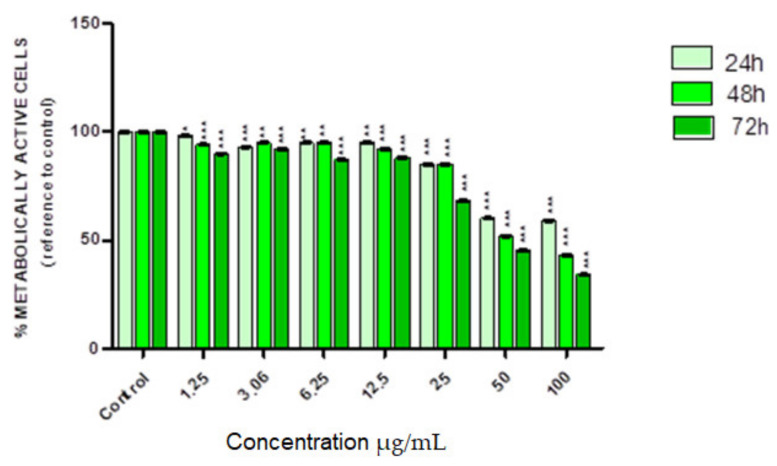
Viability of HepG2 cells after treatment with different concentrations of SPION@Au–PEG bioconjugate. The viability of cells was measured using MTS assay after incubation for 24, 48, and 72 h. The results are expressed as a percentage of control cells. Significant *p* ≤ 0.1 (*), *p* ≤ 0.01 (**), *p* ≤ 0.001 (***).

**Figure 7 ijms-24-05282-f007:**
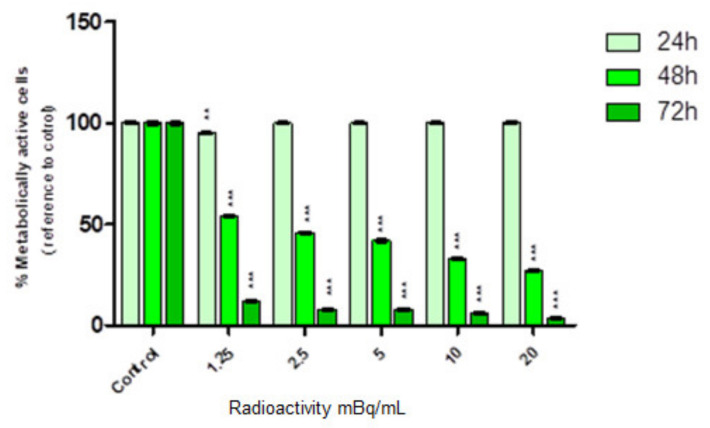
Cell viability after treatment with different radioactivity of SPION-^198^Au–PEG conjugates. HepG2 cells were incubated for 24, 48, and 72 h. The results are expressed as a percentage of control cells. Significant *p* ≤ 0.01 (**), *p* ≤ 0.001 (***).

**Table 1 ijms-24-05282-t001:** Values of average hydrodynamic diameter, zeta potential and polydispersity index (PDI) of the SPION@Au and SPION@Au–PEG nanoparticles.

Sample	Hydrodynamic Diameter (nm)	PDI	Zeta Potential (mV)
SPIONs citrate	87.68 ± 0.50	0.251	−48.5
SPION@Au	131.6 ± 3.07	0.375	−36.0
SPION@Au–PEG	99.39 ± 0.22	0.196	−39.6

## Data Availability

Not applicable.

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
