# Peer review of "198Au-Coated Superparamagnetic Iron Oxide Nanoparticles for Dual Magnetic Hyperthermia and Radionuclide Therapy of Hepatocellular Carcinoma"

_ijms, 2023, doi:10.3390/ijms24065282_

Round 1
Reviewer 1 Report (Previous Reviewer 1)
Reject
Author Response
Unfortunately, there are no objections to publication in the review. Is only a conclusion. So I can't answer.
Reviewer 2 Report (Previous Reviewer 2)
The authors responded to comments well and addressed them in detail. However, for simplicity of the readers, the authors need to label samples in the plots (Fig. 3a and 3b), and Fig.5 needs to be labeled and should be mentioned in the figure legend as well.
Author Response
The authors responded to comments well and addressed them in detail. However, for simplicity of the readers, the authors need to label samples in the plots (Fig. 3a and 3b), and Fig.5 needs to be labeled and should be mentioned in the figure legend as well.
Good remarks, the legends for the Figures have been corrected.
Reviewer 3 Report (New Reviewer)
The submitted study is aimed at relatively attractive scientific area. Any contribution to potential improvement in treatment of hepatocellular carcinoma seems to be very asked. Although the study is mainly engaged in biopharmaceutical aspects of new nanoformulation, it could subsequently serve as a basis for relevant biological experiments. However, the reviewer has several comments and recommendations:
1. The authors studied stability of the prepared nanoformulation (l. 355-360). What does it mean? The manuscript does not contain data on results of this examination. There is no information on radiochemical evaluation (by HPLC or ITLC) of the prepared radiolabeled conjugate. Therefore, no data on potential release of 198Au from the nanoparticles are included even this parameter is very important to characterize stability of the tested radioconjugate.
2. Figure 7: Which concentration of the tested conjugate was used? Did you use several preparations with the same concentration of nanoparticles and various radioactivity? Or did you use one preparation and add various amount of the nanoparticles (and radioactivity) to cells? Information on exact procedure is missing in Methods.
3. Figure 2: Is it possible to provide more clearer microscopic pictures of the nanoformulations? Quality of the presented images is not high.
4. Figure 5: The symbols a, b, c presented in the figure are not fit to relevant formulations presented in the legend. In addition, these letters in the figure are too small for reading.
5. The third paragraph on page 5 (l. 190-201) describes procedure how to determine SAR. But the procedure is presented in Results and Discussion. This text should be moved to the relevant location in the methodological part.
6. Abstract: The authors should consider whether they should rather use the past tense in the results of the abstract.
7. No list of abbreviations is included. Therefore, all abbreviations in figures and tables should be explained in their legends (PDI in Table 1; H in Fig. 3).
Author Response
- The authors studied stability of the prepared nanoformulation (l. 355-360). What does it mean? The manuscript does not contain data on results of this examination. There is no information on radiochemical evaluation (by HPLC or ITLC) of the prepared radiolabeled conjugate. Therefore, no data on potential release of 198Au from the nanoparticles are included even this parameter is very important to characterize stability of the tested radioconjugate.
As mentioned in our publication, we measured the stability of our nanoconjugates in terms of colloidal stability and non-agglomerate formation. As is well known due to the noble nature of metallic gold, the release of Au in ionic form is not observed. This sentence has been added to the text.
2. Figure 7: Which concentration of the tested conjugate was used? Did you use several preparations with the same concentration of nanoparticles and various radioactivity? Or did you use one preparation and add various amount of the nanoparticles (and radioactivity) to cells? Information on exact procedure is missing in Methods.
This is a very important remark. In the case of radioactive nanoparticles, the radioactivity of 198Au administered to the cells was regulated by changing the concentration of the 198Au-PEG conjugate. We prepared a solution of 198Au-PEG with a concentration of 1 MBq/ug and by increasing the concentration we obtained solutions with higher radioactivity concentrations, up to 20 MBq/ml. No cytotoxicity was observed in this concentration range for non-radioactive conjugates tested.
3. Figure 2: Is it possible to provide more clearer microscopic pictures of the nanoformulations? Quality of the presented images is not high.
Unfortunately, we couldn't get better images on our device.
4. Figure 5: The symbols a, b, c presented in the figure are not fit to relevant formulations presented in the legend. In addition, these letters in the figure are too small for reading.
We changed the figures, captions and added symbols to the text.
5. The third paragraph on page 5 (l. 190-201) describes procedure how to determine SAR. But the procedure is presented in Results and Discussion. This text should be moved to the relevant location in the methodological part.
Done, thank you
6. Abstract: The authors should consider whether they should rather use the past tense in the results of the abstract.
Done, thank you.
7. No list of abbreviations is included. Therefore, all abbreviations in figures and tables should be explained in their legends (PDI in Table 1; H in Fig. 3).
Done, thank you
This manuscript is a resubmission of an earlier submission. The following is a list of the peer review reports and author responses from that submission.
Round 1
Reviewer 1 Report
In this work, authors synthesized the magnetite nanoparticles and covered them with a layer of radioactive 198Au creating core-shell nanoparticles. The synthesized SPION@Au nanoparticles exhibit superparamagnetic properties with saturation magnetization of 50 emu/g, which is lower than reported for uncoated SPIONs (83 emu/g). the work looks interesting and after addressing the following comment can be considered for further evaluation
-The figures at line 210 don’t have a caption, please check and revise
-Please add “significant difference” with p-value to figures 4 and 5.
-it is suggested to add the following studies to your paper and discuss them to enrich your paper: https://doi.org/10.2147/IJN.S243223, https://doi.org/10.1073/pnas.2004121117, https://doi.org/10.1002/advs.202003535, https://iopscience.iop.org/article/10.1088/1361-6528/abf878/meta
-it is better to add a graphical abstract to present the goal of the study
-the conclusion is too lengthy and does not provide a solid one, please summarize and revise it
Author Response
-The figures at line 210 don’t have a caption, please check and revise
Done, thank you
-Please add “significant difference” with p-value to figures 4 and 5.
-it is suggested to add the following studies to your paper and discuss them to enrich your paper: https://doi.org/10.2147/IJN.S243223, https://doi.org/10.1073/pnas.2004121117, https://doi.org/10.1002/advs.202003535, https://iopscience.iop.org/article/10.1088/1361-6528/abf878/meta
These references are related to topics unrelated to the work presented.
1. doi is incorrectly specified
2. The publication is related to the study of MOF using 129XE NMR. It does not concern the issue of our publication.
3. The work is about a leukemia inhibitor, also not related to our work
4. The work concerns CoFe2O4 with folic acid as a doxorubicin carrier. Subject of publication is not related to the application of radionuclides or HCC cancer therapy presented in our work.
-it is better to add a graphical abstract to present the goal of the study
Done. The graph is at the end of the Introduction
-the conclusion is too lengthy and does not provide a solid one, please summarize and revise it
Done. Conclusions have now only 17 lines.
Reviewer 2 Report
The authors have studied dual magnetic hyperthermia and radionuclide therapy for surface-functionalized SPIONs in HepG2 cells. The citrate-coated superparamagnetic iron oxide-based nanoparticles (SPIONs) are functionalized with radioactive 198Au and/or polyethylene glycol (PEG) to create core-shell nanoparticles. Although the core-shell magnetite-Au nanoparticles have less saturation magnetization than uncoated SPIONs, they can reach hyperthermic temperature (43°C) after inducing with a magnetic field frequency of 386 kHz. The physio-chemical characterization and in-vitro experiments are performed, and it is well-written and described in the manuscript; however, a few minor comments must be addressed.
Comments.
1. In the table. 1. dynamic light scattering (DLS) measurement of SPION citrate hydrodynamic size and zeta potential should be included.
2. The DLS measurements for size distribution and zeta potential of all three-surface functionalized SPIONs need to represent in the graph.
3. In figure.2. To confirm the superparamagnetic property of SPION@Au, zoom-in of the hysteresis at 0 magnetization v/s H [Oe] needs to show as an insert or in an additional plot.
4. For the plot’s temperature change v/s time, figure legends and captions are missing. I believe Figure 4 is meant to be figure 5, like figure 5 to figure 6.
5. To confirm whether the synthesized SPIONS@Au (since it has low magnetic saturation compared to SPION citrate) has a cytotoxicity effect from reactive oxygen species (ROS) from the Fenton reaction mechanism. Therefore, the ROS detection assay must be performed and included in cell cytotoxicity studies.
6. In the introduction, while discussing SIPONs, it could be included that the surface functionalized magnetic nanoparticles’ simultaneous induction of alternating magnetic field and Near-Infrared laser can increase hyperthermic temperature (43°C) at lower concentrations and could cite the following two references. “Int. J. Mol. Sci. 2020, 21(15), 5187; https://doi.org/10.3390/ijms21155187 and Micromachines 2022, 13(8), 1279; https://doi.org/10.3390/mi13081279”.
7. Another minor notice: In the introduction, line 66, the sentence period is used before and after the reference. In section 3.4, line 231, “conducted on da by initial treatment,” the underlined is unclear.
Author Response
In the table. 1. dynamic light scattering (DLS) measurement of SPION citrate hydrodynamic size and zeta potential should be included.
Done
- The DLS measurements for size distribution and zeta potential of all three-surface functionalized SPIONs need to represent in the graph.
We believe that we should not present the same data in a table and graphically. Typically, DLS measurements are presented in a table. It illustrates them well.
In figure.2. To confirm the superparamagnetic property of SPION@Au, zoom-in of the hysteresis at 0 magnetization v/s H [Oe] needs to show as an insert or in an additional plot.
- Done, good remark, thank you.
- For the plot’s temperature change v/s time, figure legends and captions are missing. I believe Figure 4 is meant to be figure 5, like figure 5 to figure 6.
Done, thank you
- To confirm whether the synthesized SPIONS@Au (since it has low magnetic saturation compared to SPION citrate) has a cytotoxicity effect from reactive oxygen species (ROS) from the Fenton reaction mechanism. Therefore, the ROS detection assay must be performed and included in cell cytotoxicity studies.
Unfortunately, we do not have the possibility to conduct an experiment on ROS determination. However, it is known from the literature that Fe3O4 can generate small amounts of ROS. (The role of ROS generation from magnetic nanoparticles in an alternating magnetic field on cytotoxicity, Acta Biomater. 2015, 25:284-90.) Since the Au layering of SPION is not complete, SPION@Au nanoparticles can also generate ROS.
- In the introduction, while discussing SIPONs, it could be included that the surface functionalized magnetic nanoparticles’ simultaneous induction of alternating magnetic field and Near-Infrared laser can increase hyperthermic temperature (43°C) at lower concentrations and could cite the following two references. “Int. J. Mol. Sci. 2020, 21(15), 5187; https://doi.org/10.3390/ijms21155187 and Micromachines 2022, 13(8), 1279; https://doi.org/10.3390/mi13081279”.
Done, we added
-Another minor notice: In the introduction, line 66, the sentence period is used before and after the reference. In section 3.4, line 231, “conducted on da by initial treatment,” the underlined is unclear.
Done, thank you.
Round 2
Reviewer 1 Report
Accept in present form